# The causal role of affect sharing in driving vicarious fear learning

**Alexa Müllner-Huber**[1]*, **Lisa Anton-Boicuk**[1], **Ekaterina Pronizius**[1], **Lukas Lengersdorff**[1], **Andreas Olsson**[2], **Claus Lamm**[1]*

1 Social, Cognitive and Affective Neuroscience Unit, Department of Cognition, Emotion, and Methods in Psychology, University of Vienna, Vienna, Austria, 2 Division of Psychology, Department of Clinical Neuroscience, Karolinska Institute, Stockholm, Sweden

* alexa.huber@univie.ac.at (AMH); claus.lamm@univie.ac.at (CL)

**Data Availability Statement:** All relevant data are within the paper and its Supporting Information files.

## Abstract

Vicarious learning, i.e. learning through observing others rather than through one's own experiences, is an integral skill of social species. The aim of this study was to assess the causal role of affect sharing, an important aspect of empathy, in vicarious fear learning. N = 39 participants completed a vicarious Pavlovian fear conditioning paradigm. In the learning stage, they watched another person–the demonstrator–responding with distress when receiving electric shocks to a color cue (conditioned stimulus; CS+; a different color served as CS-). In the subsequent test stage, an increased skin conductance response (SCR) to the CS+ presented in the absence of the demonstrator indexed vicarious fear learning. Each participant completed this paradigm under two different hypnotic suggestions, which were administered to induce high or low affect sharing with the demonstrator in the learning stage, following a counterbalanced within-subject design. In the learning stage, high affect sharing resulted in stronger unconditioned SCR, increased eye gaze toward the demonstrator's face, and higher self-reported unpleasantness while witnessing the demonstrator's distress. In the test stage, participants showed a stronger conditioned fear response (SCR) when they had learned under high, compared to low, affect sharing. In contrast, participants' declarative memory of how many shocks the demonstrator had received with each cue was not influenced by the affect sharing manipulation. These findings demonstrate that affect sharing is involved in enhancing vicarious fear learning, and thus advance our understanding of the role of empathy, and more generally emotion, in social observational learning.

## Introduction

In social species such as our own, fears of and knowledge about what is dangerous and should be avoided are often learned when we observe others in distressing situations [1]. This phenomenon of vicarious fear learning has been extensively studied using the vicarious Pavlovian fear conditioning paradigm [2], in which human participants learn to fear a visual cue by watching videos of another person ("the demonstrator") receive painful electric shocks following that cue.

**Funding:** Open access funding provided by University of Vienna.

**Competing interests:** The authors have declared that no competing interests exist.

While an essential role of social cognitive skills for such learning via others' experiences seems intuitive, definitive insights into how and what aspects of social cognition support vicarious fear learning are still missing [3]. One mechanism that has been hypothesized to play a critical role is affect sharing [4–6], a core aspect of empathy that describes the ability to partially re-experience how another person is feeling [7]. The few existing studies connecting empathy to vicarious fear learning in humans have been correlational and have obtained somewhat inconsistent results [8–10]. Initial support for a more causal role of empathy came from a recent experiment in humans. Olsson et al. [9] observed higher learned fear responses in participants instructed to pay attention to the demonstrator's distress, and told that the shocks were actually painful, compared to another group informed that the demonstrator was an actor faking pain. However, these instructions targeted more cognitive aspects of empathy such as mentalizing, i.e. thinking about rather than sharing the demonstrator's feelings [11]. In addition, telling participants that the shocks were painful may have increased their threat value and, thus, the learned fear response, directly, without requiring affect sharing.

There is thus a lack of evidence on the specific involvement of affect sharing in vicarious fear learning.

To close this gap, we employed hypnosis as an experimental tool to manipulate affect sharing more specifically. Hypnosis is highly effective in manipulating pain and emotions [12–14] and has also been used to manipulate emotional responses to others [15, 16]. In the current study, we used hypnotic suggestions to induce either high or low affect sharing with the demonstrator in the learning stage of the vicarious fear conditioning paradigm. We predicted that both affect sharing and fear learning, as indicated by self-report and skin conductance response (SCR), would be higher following suggestions for high as compared to low affect sharing. Possible longer-lasting changes in sympathetic arousal were also assessed by measuring tonic skin conductance level (SCL).

Although our hypnotic suggestions targeted affect sharing specifically, they may have (additionally) triggered more cognitive emotion regulation strategies involving, e.g., mentalizing or visual attention [17]. To assess the occurrence of such cognitive mechanisms, which may also affect vicarious fear learning, qualitative interviews were conducted at the end of each experiment. To explore the role of visual attention, eye gaze was analyzed for whether participants spent more time looking at the demonstrator's face during high as compared to low affect sharing, and whether this correlated with fear learning.

Finally, as participants may learn the cue-shock contingency even if they are not afraid of the shocks, declarative memory of the cue-shock contingency was tested at the end of the experiment as an alternative learning measure less reliant on fear. We expected better performance under high compared to low affect sharing, based on the assumption that high affect sharing should foster associative learning of the cue-shock contingency.

## Materials and methods

All experimental procedures were approved by the ethics committee of the Medical University of Vienna (ethics clearance no. 2019/2017) and were conducted in accordance with the World Medical Association declaration of Helsinki, current revision (October 2013).

### Sample and screening

N = 410 first- and second-year Bachelor Psychology students of the University of Vienna were screened for hypnotic suggestibility using the Harvard Group Scale of Hypnotic Susceptibility (HGSHS:A) [18]. Participants were recruited from this pool if they had a hypnotic suggestibility score of at least 7 on a scale ranging from 0–12, in line with previous hypnosis studies, e.g.

[12]. This criterion was met by approximately one third of the student population. Additional exclusion criteria were a history of chronic pain, acute pain on the day of the experiment, previous experience with experimental electrical pain stimuli, a history of substance abuse or psychiatric or severe organic disease, and chronic or recent use of opiates.

Based on the medium effect size of Cohen's d = 0.50 observed in Olsson et al. [9], we conducted an *a priori* power analysis to detect a medium effect size (Cohen's f = 0.25) with the conventional power of 80% for our central effect of interest, i.e. the CS-by-suggestion interaction observed in a 3-way mixed ANOVA including the repeated-measures factors CS and suggestion and the between-subjects factor presentation order. This yielded a minimum sample size of N = 34 participants using a within-subject design. Due to delays in the technical data inspection, data collection was terminated once we had collected 36 participants with technically valid data on all measures. A total of 42 students were recruited for the experiment, of which three were excluded from the analysis due to movement artefacts in the SCR signal (N = 2) or because they decided to terminate participation due to a negative emotional reaction during the experiment (N = 1). Another three participants were excluded from the analysis of eye gaze data due to missing gaze data (N = 2; see details below) or technical problems with the eye tracker (N = 1), resulting in a final sample size of 39 participants (30 female, 9 male; mean age 20.0 years, range 18–24 years) for self-report and SCR, and 36 participants (28 female, 8 male) for the eye gaze results. Participants provided written informed consent to participate in the study and were compensated with student credits for the screening and with 45 € for participation in the experiment.

## Vicarious fear conditioning paradigm

A summary of the experimental design and procedure is given in Fig 1. The differential Pavlovian fear conditioning paradigm was implemented in line with the protocol of [2], and it consisted of a vicarious learning stage followed by a test stage. Using a within-subjects design aimed at maximizing effect sizes, each participant completed the paradigm under two different hypnosis conditions (2 rounds, see below) inducing high or low affect sharing in the learning stage. Presentation order of the hypnotic conditions was counterbalanced across participants–19 participants received the high affect sharing condition in round 1 and the low affect sharing condition in round 2 (group 1), while 20 participants received the reversed order (group 2). All other aspects of the paradigm, including all information given about the demonstrator, were kept constant. Any differences observed in the learned fear response would therefore demonstrate a role of affect sharing in modulating vicarious fear learning.

## Learning stage

During each vicarious learning stage, the participant watched a series of 12 pre-recorded video clips showing a male demonstrator in a setting similar to their own. Two differently colored squares (cues) were presented on the demonstrator's monitor six times each in quasi-randomized order. A different demonstrator and cue color pair was used in the round 2 of the paradigm to avoid carry-over effects on learning from round 1 (both counterbalanced across participants)–see Fig 1. The used cue colors were green and pink in round 1, and blue and yellow in round 2 of the paradigm, with order reversed for half of the participants. One of the colored squares served as the conditioned stimulus (CS+), while the other one served as the control stimulus (CS-); this assignment was counterbalanced across participants. Four out of the 6 presentations of the CS+ ended with an electric shock being delivered to the demonstrator's right arm, to which the demonstrator responded with a visible facial expression of pain and a slight jerk of the right arm (unconditioned stimulus, US). The CS- was never followed by

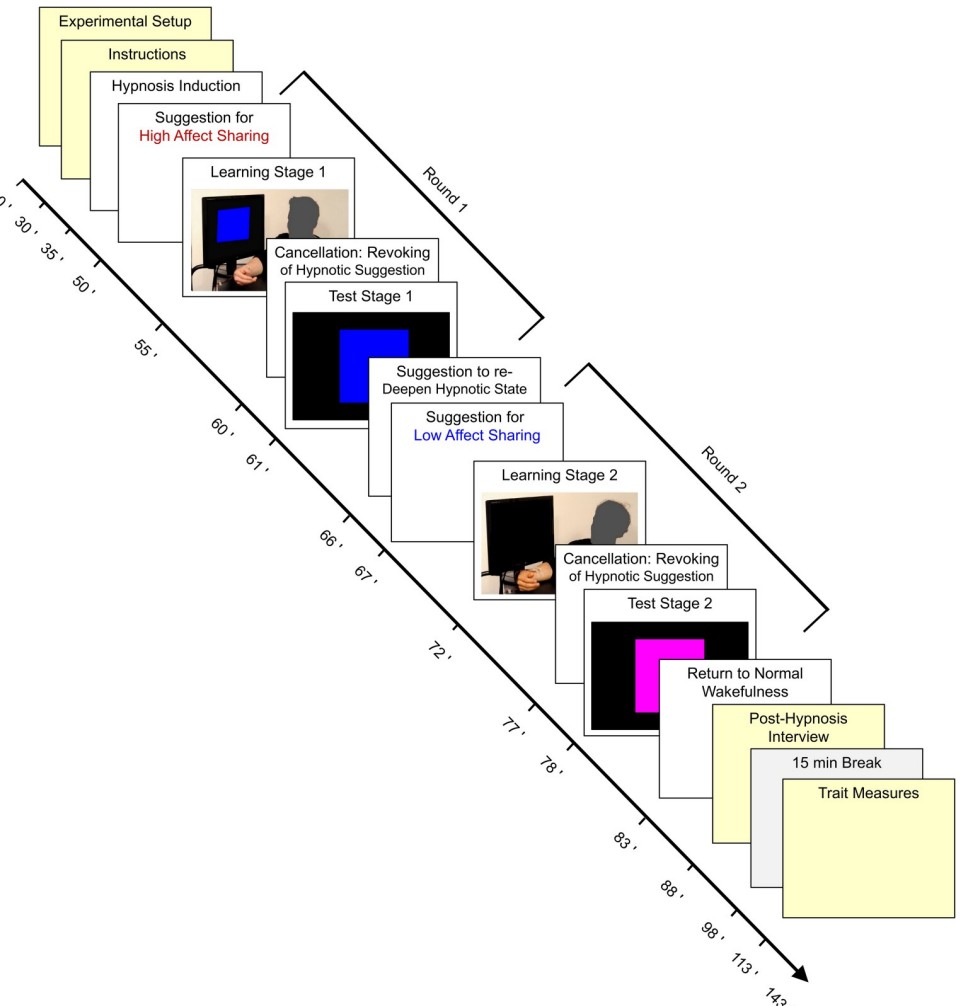

**Fig 1. Overview of the experimental design and procedures.** The vicarious fear conditioning paradigm consisted of a learning stage, in which the participant watched videos of a demonstrator receiving electric shocks paired with a predictive visual cue, followed by a test stage, in which an increased skin conductance response (SCR) to the cue compared to a control stimulus, in the absence of the demonstrator, indexed vicarious fear learning. Each participant completed this paradigm under two different hypnotic suggestions, which were administered before each learning stage to induce high or low affect sharing with the demonstrator (order counterbalanced across participants). Each suggestion was revoked (so-called "cancellation") after the learning stage to bring the participants back to their habitual level of affect sharing before the start of the test stage. Eye tracking was used to assess eye gaze during the learning stages. The demonstrators' heads have been masked in this Figure to protect their identity; participants saw their full face and expressions.

a shock to the demonstrator (the demonstrator's calm face following a CS- signaled the absence of the US). The videos are a well-tested stimulus set, which has been successfully used in previous studies on vicarious fear conditioning [2].

Video clips were presented at approximately 23.8 x 13.6˚ visual angle. Each video clip (trial) had a total duration of 9 s and started with the presentation of the square for 6 s. Electric shocks were delivered to the demonstrator 5.5 sec after trial start and lasted for 100 ms, while the demonstrator's facial response lasted for approximately 2 s. The single 9 s long video clips were divided by an inter-trial-interval (ITI) showing a white fixation cross on black back-ground, jittered to range from 8–12 s duration, resulting in a total trial duration of 19 s on average.

### Test stage

During the test stage, the same colored squares shown in the videos of the preceding learning stage were now presented on the participant's computer monitor, 6 times the CS+ and 6 times the CS- in quasi-randomized order. No demonstrator was shown, and SCR upon cue presentation was recorded as an index of fear learning. In line with standard procedures [2] aimed at ensuring that only indirect, vicarious learning is measured, the participant never actually received any shocks during the entire test stage. Each trial started with a 6 s presentation of the colored square, followed by an ITI of 13 s duration on average (jittered between 11–15 s). For further details see [2].

### Procedure and instructions

The procedure is summarized in Fig 1. Upon signing the informed consent form, participants were informed that they might receive between 0 and 4 uncomfortable but not painful electric shocks delivered to the right hand during each of the two test stages of the hypnosis session [2]. Importantly, participants were not given any information as to which cue(s) would be followed by shocks. In contrast to Haaker et al. [2], participants were told that they could *not* receive any shocks during the learning stages. This served to allow them to fully concentrate on the hypnotic suggestions without getting distracted by fear of shocks. The information that they might receive between 0 and 4 shocks in each test stage was adapted from Olsson et al. [9] and served as a cover story to make participants still expect shocks in the second test stage after not receiving any in the first one. After the second round of the vicarious fear conditioning paradigm, participants were guided back to the normal waking state and completed the post-hypnosis interview. Participants then went outside the laboratory for a 15-min break, after which they filled in several trait questionnaires, were debriefed, compensated for their time, and released. See S1 File for further details on the instructions and on the trait measures, and Table G in S1 File for the average results on those trait measures.

### Hypnotic induction and suggestions

All hypnotic procedures were delivered by the hypnotist and co-author L.A.B., a trained clinical hypnotist, sitting next to the participant and following a standardized protocol. The hypnotic state was induced using a 15-min protocol similar to standard induction procedures [18], which included suggestions for focused attention, relaxation, eyes closure and finally the suggestion that, afterwards, upon reopening their eyes, the participants would be able to perform the task fully concentrated while remaining in the hypnotic state. After having completed the first learning and test stage, participants closed their eyes again and received a brief suggestion to renew / deepen their hypnotic state, in line with previous studies [12].

The suggestion for increased affect sharing included suggestions to be open and sensitive for the feelings of others and to feel what the demonstrator feels (". . .it is like an open window through which all emotions can flow freely between two people. . ."). Upon completion of the learning stage, the so-called suggestion cancellation was delivered by saying: "You are now again as open as you were at the beginning of this session. . ." to bring the participant back to their normal level of affect sharing. Following a brief reminder presented on screen that they would now complete the same experiment as the person in the videos, and that they could receive 0–4 electric shocks now, participants completed the test stage. The suggestion for reduced affect sharing included suggestions to be closed against and distanced from the feelings of others, and not to feel what the demonstrator feels (". . .it is like a thick, bullet-proof glass that protects you against other people's emotions. . ."). The procedure was otherwise identical to the suggestion for increased affect sharing.

## Physiological measures

Participants were seated on a recliner, with a 24-inch computer monitor placed before them. A Biopac (Biopac Systems Inc., Goleta CA, USA) bar electrode (filled with electrode gel) similar to the one shown in the video clips was attached to the wrist of the participant's dominant right hand. It served ostensibly to deliver unpleasant electric shocks during the test stages. However, the participant never actually received any shocks during the entire experiment. Electrocardiogram (ECG, outside scope of present paper), skin conductance and eye movements were recorded throughout the paradigm. Skin conductance level was recorded using an 8-channel bioamplifier (Mobi8-BP; TMSi B. V., Enschede, the Netherlands), with a custom-specific skin-conductance sensor consisting of two reusable flat Ag/AgCl electrodes placed at the medial phalanges of the index and middle finger of the left hand without gel and fixated with velcro straps, in strict adherence to manufacturer instructions.

Eye movements of the participant's dominant eye were recorded with an EyeLink 1000 Plus Desktop Mount Eye Tracker (SR Research Ltd., Kanata, Ontario, Canada) at a sampling rate of 500 Hz (eye-to-screen distance 1 m, camera-to-screen distance 375 mm) without chin rest using a 16 mm lens and a target sticker placed on the forehead above the dominant eye, which allowed the participants to move their head freely, making them more comfortable during the hypnosis session. The eye tracker was calibrated using a 13-point calibration procedure at the beginning of each learning and test stage, respectively. After trials 4 and 8 of each learning and test stage, respectively, a drift check was performed, which required participants to fixate on a centrally presented target cross. The drift check was repeated, if necessary, until recorded fixation gaze was inside of a 4˚ radius of the target cross.

## Post-hypnosis interview

Upon completion of the hypnosis session, participants immediately answered a series of questions presented on screen. Among other things, they estimated how many electric shocks the demonstrator had received with each color, using a 7-point numerical rating scale ranging from 0 (none) to 6+ (6 or more). They rated how unpleasant the shocks were for the demonstrator, how unpleasant it was for them to watch these videos, and how likeable they found the demonstrator. They also indicated whether or not they had felt any tingling and / or pain in their own body while watching the demonstrator receive electric shocks (yes/no response). Finally, they answered the following open-ended questions, separately for each suggestion: What strategy did you use to implement the suggestion? What changes did the suggestion induce while you watched the video? See S1 File for a complete description of the post-hypnosis interview.

## Data analysis

**Self-report.** To assess (a) which strategies were used by the participants to implement the hypnotic suggestions, and (b) what effects these suggestions had on their subjective experience during the experiment, answers given to the last 2 questions of the post-hypnosis interview (see above and S1 File) were categorized into a system of 11 categories including the emotion regulation strategies described in [17] and several categories added ad-hoc based on the interview data. Single participants could give answers falling into more than one category, and some participants gave no answers at all. The categories and their results are shown in Table A in S1 File.

**Skin conductance.** Skin conductance data were sampled at 256 Hz and subsequently preprocessed and analyzed using MATLAB R2017b (The MathWorks, Inc., Natick, MA) and Ledalab V3.4.9 (Leipzig, Germany). Data were inspected for possible movement-related artefacts, low-pass filtered using a 5 Hz threshold and down-sampled to 16 Hz. Continuous

decomposition analysis (CDA) based on standard deconvolution was performed to divide the signal into two distinct continuous measures reflecting tonic and phasic electrodermal activity, respectively. This approach is more sensitive and more robust against artefacts compared to traditional trough-to-peak analysis [19]. SCR amplitude was calculated as the sum of amplitudes of all reconvolved phasic SCRs with onset occurring within a time window of 0.5–4.5 sec after CS/US onset, using a minimal threshold of 0.02 µS, in line with previous studies [2]. Responses that did not pass this threshold were scored as 0. Raw values were logarithmized to improve normality of distribution, and z-transformed within each participant for standardization to render data comparable between participants [20–22]. For the analysis of correlations between learning and test stage across participants, the z-transformation was calculated separately for each stage and suggestion in order to avoid the introduction of spurious anti-correlations associated with global signal removal [22, 23]. In line with previous studies [8, 9], the first trial of each condition (colored squares: CS+, CS-; demonstrator's response: US, US absence) was excluded from the analysis, because the SCR was expected to be unusually large in these trials due to the newness of the situation, introducing error variance to the results. The difference in average SCR induced by the US (following a CS+) versus the US absence (following a CS-) in the learning stage is termed the unconditioned response (UR) (i.e., the response specifically induced by watching the demonstrator in pain). Note that the two trials in which a US absence followed a CS+ were excluded from this US analysis because they contained an element of surprise. The difference in average SCR between the CS+ and the CS- was calculated for the learning and test stage, respectively, and is termed the conditioned response (CR), reflecting the degree to which fear learning had happened.

To detect possible changes in more slowly varying tonic arousal level across the 2 learning and test stages of the paradigm, the CDA-based tonic skin conductance level (SCL) signal was logarithmized, averaged across all trials of each stage, using the entire 9-sec trial duration as time window, and z-transformed within each participant. To make results more comparable to the SCR data, the first trial of each condition (CS+, CS-) was excluded from the analysis. SCL is thought to reflect sympathetic nervous system arousal [22].

**Eye tracking.** The Eyelink 1000 Plus online parser was used with default settings to extract fixations, saccades and blinks from the eye movement data ("cognitive configuration" for Remote Mode: use gaze data to compute velocity; velocity threshold of saccade detection: 40˚/sec; acceleration threshold of saccade detector: 80000˚/sec/sec; minimum motion out of fixation before saccade onset allowed: 0.2˚; maximum pursuit velocity accommodation by the saccade detector: 60˚/sec; fixation update interval: 50 msec). These data were further analyzed using EyeLink Data Viewer 3.1.97 software (Mississauga, Ontario, Canada: SR Research Ltd, 2017). Gaze data were extracted during the CS+/CS- events (time period between 0 and 5500 ms after CS onset) and the US/US absence events (time period between 5500 and 9000 ms after CS onset, i.e. from US onset to trial end), respectively, of each learning stage. Two participants were excluded due to missing gaze data using the criterion described in [8], i.e., that at least one event contributed < 1000 ms (for CS events) or < 600 ms (for US events) valid fixation time at the screen.

Two areas of interest (AOI) were defined for each video series: The colored square / CS (termed "cue") presented to the demonstrator (AOI1) and the demonstrator's face (AOI2). The average fixation time per trial within each of the two AOI was calculated for each participant, expressed as % of total fixation time per trial.

## Statistical analysis

The effects of hypnotic suggestions on self-report, SCR and eye gaze were assessed using a mixed-model analysis of variance (ANOVA). Note that in a previous study, completing the

vicarious fear conditioning paradigm twice such as here did not significantly affect the results [9]. However, as SCR were generally smaller in round 2 compared to round 1 of the paradigm, in keeping with habituation effects documented in the SCR literature [21, 22], presentation order (suggestion for high affect sharing presented in round 1 or in round 2, see Fig 1) was included as a between-subjects factor of no interest in all analyses to remove error variance produced by this effect. 95% confidence intervals shown in line plots depicting repeated-measures designs were calculated following [24] using an SPSS macro described in [25]. This allows to visually interpret confidence intervals based on dependent samples in the same way as confidence intervals from independent samples [24].

Possible correlations between SCR and eye gaze across participants were assessed using Spearman correlations. To control for order, all experimental variables were residualized of order using multiple regression before conducting these correlations. To avoid type I error inflation caused by multiple redundant eye gaze variables, a composite eye gaze index was calculated by averaging the % fixation time at the demonstrator's face versus the cue across all CS and US events, separately for each suggestion condition. Only this composite index, which represents the only significant change in eye gaze induced by suggestions (see Results section and Table E in S1 File) was included in the correlation analysis, to assess whether the magnitude of changes in eye gaze correlated with SCR measures.

To assess whether correlations between self- and other-experienced unpleasantness ratings from the post-hypnosis interview were modulated by the hypnotic suggestions, and to compare the correlations of different learning stage variables with the learned fear response in the test stage, we tested for significant differences in correlations, using the test procedures recommended by [26] to compare dependent correlation coefficients, as implemented in the R package *psych*, function *r.test*.

## Results

### Test stage: Skin conductance response

We first assessed the effect of learning under hypnotic suggestions for high versus low affect sharing on the conditioned fear response observed in the test stage. To this end, a 3-way mixed ANOVA with suggestion order (high or low affect sharing first) as between-subjects factor and suggestion (high / low affect sharing) and CS (the two colored squares) as within-subject factors was calculated, using SCR upon cue presentation in the test stage as the dependent variable. Results demonstrated that the conditioned fear response (CR), indicated by a stronger SCR to the CS+ compared to the CS-, was significantly stronger when learning had occurred under hypnotically induced high as compared to low affect sharing (suggestion x CS interaction, $F(1, 37) = 5.68$, $p = .022$, $\eta_p^2 = 0.13$, on top of a CS main effect, $F(1, 37) = 21.26$, $p < .001$, $\eta_p^2 = 0.36$). See Fig 2C for illustration. Post-hoc 2-way (CS x order) ANOVAs calculated separately for each affect sharing condition confirmed that participants showed a stronger SCR to CS+ compared to CS- only in the high affect sharing ($F(1,37) = 21.33$, $p < .001$, $\eta_p^2 = 0.37$), not in the low affect sharing condition ($F(1,37) = 2.58$, $p = .117$, $\eta_p^2 = 0.07$).

Differences between the two order groups: There was also a significant suggestion x order interaction ($F(1,37) = 63.77$, $p < .001$, $\eta_p^2 = 0.63$). Order is a between-subjects factor dividing the whole sample into two groups based on the order in which participants received the two suggestions (group 1 received high affect sharing in round 1, group 2 received it in round 2). This means that possibly occurring general habituation effects (round 1 > round 2) will always boost the suggestion effect (high > low affect sharing) in group 1, but will diminish it in group 2. In other words, here and throughout the results section, a significant suggestion x order

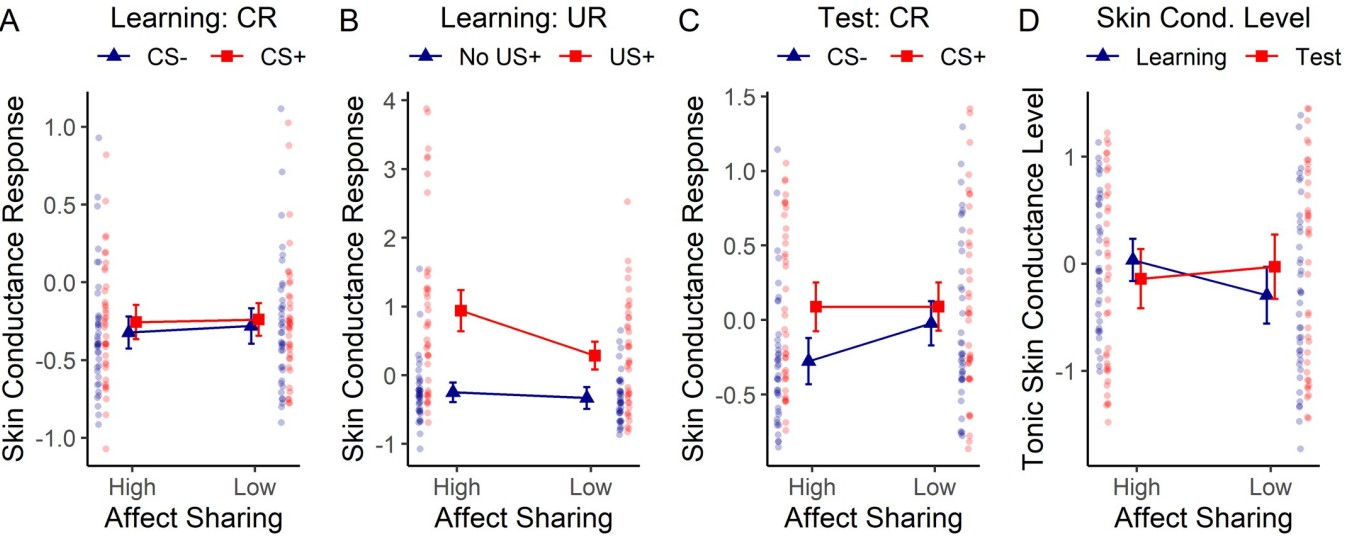

**Fig 2.** Effects of hypnotic suggestions for high versus low affect sharing on skin conductance response (SCR) in the learning (A, B) and test stage (C) and on tonic skin conductance level (SCL) across both stages (D), expressed as z-scores. A) SCR to seeing the colored square (CS+ or CS-) in the learning stage. B) SCR to seeing the demonstrator receive shocks (US) or no shocks (US absence) in the learning stage. The difference in responses to US and US absence indicates the unconditioned response. C) SCR to seeing the colored square (CS+ or CS-) in the test stage. The difference in responses to CS+ and CS- indicates the conditioned response. D) Tonic skin conductance level (SCL) observed in the learning and test stage. Error bars reflect 95% confidence intervals corrected for within-subject designs (see Methods). Results of individual participants are shown laterally as semi-transparent dots.

interaction always demonstrates differences between round 1 and 2. Here the result indicates that SCR were generally lower in the second compared to the first test stage (compare Fig 3C). All other effects of this ANOVA model were non-significant–see Table C in S1 File for the complete results.

### Learning stage

**Self-report.** Results of the post-hypnosis interview regarding strategies used and effects obtained from the hypnotic suggestions are summarized in Table A in S1 File. For both suggestions, the strategy most frequently reported to have been used to implement the suggestion was to remember or imagine scenes in which the participant felt emotionally open (high affect sharing) versus closed (low affect sharing) towards another person (62% / 67% of participants under high / low affect sharing), and the most frequently mentioned effect of the suggestion while watching the video was to feel more or less affect sharing with the demonstrator (64% of participants), supporting the internal validity of our experimental manipulation. Cognitive strategies such as mentalizing or visual attention were mentioned as a strategy by up to 15% of participants, and as an effect of the suggestion by up to 26%–see Table A in S1 File.

In their quantitative ratings, participants described both hypnotic suggestions as equally "effective" (see Table B in S1 File). Participants' quantitative ratings on how unpleasant the shocks were for the demonstrator and how unpleasant it was for them to watch this, respectively, indicated that both other-related and self-related unpleasantness ratings were increased in the high as compared to the low affect sharing condition. However, the effect was stronger for the self-related ratings, as shown in Fig 4A (3-way order x suggestion x target ANOVA, significant effects for suggestion, $F(1, 37) = 67.72$, $p < .001$, $\eta_p^2 = 0.65$, for target, $F(1, 37) = 53.39$, $p < .001$, $\eta_p^2 = 0.59$, and a suggestion x target interaction, $F(1, 37) = 39.28$, $p < .001$, $\eta_p^2 = 0.52$).

The above main effect of suggestion was stronger in the group receiving high affect sharing in round 1 (suggestion-by-order interaction: $F(1,37) = 4.34$, $p = .044$, $\eta_p^2 = 0.11$), indicating

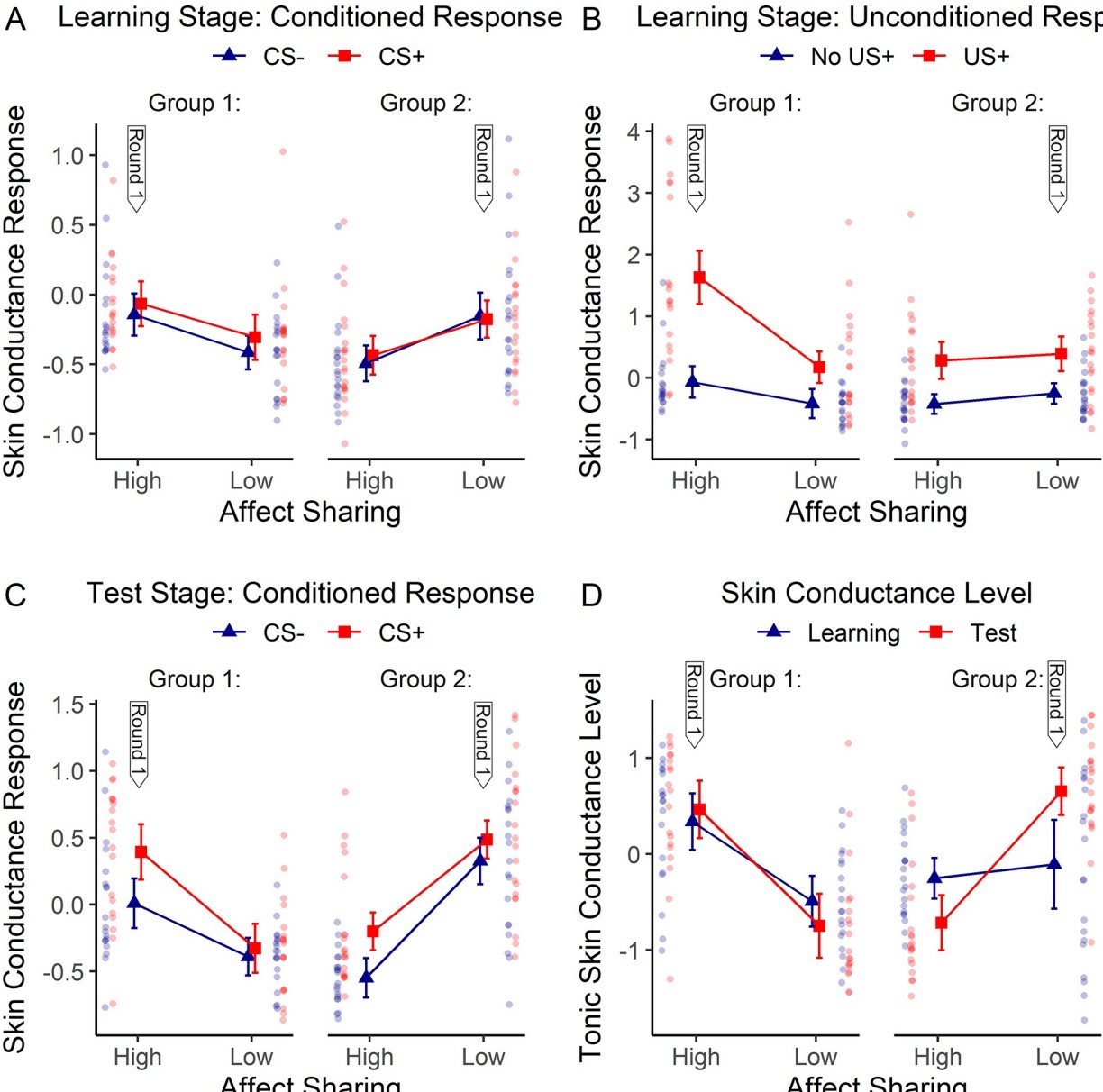

**Fig 3.** Effects of hypnotic suggestions for high versus low affect sharing on skin conductance response (SCR) (panel A, B, C) and tonic skin conductance level (SCL) (panel D), shown separately for the two groups receiving the high affect sharing condition in round 1 (group 1) or in round 2 (group 2). The thick vertical arrows ("Round 1") indicate which suggestion was delivered first in this group. Results are expressed as z-scores. A) SCR to seeing the colored square (CS+ or CS-) in the learning stage. B) SCR to seeing the demonstrator receive shocks (US+) or no shocks (No US+) in the learning stage. C) SCR to seeing the colored square (CS+ or CS-) in the test stage. D) Tonic SCL in the learning and test stage. Error bars reflect 95% confidence intervals corrected for within-subject designs (see Methods). Results of individual participants are shown laterally as semi-transparent dots.

generally lower unpleasantness ratings in the second compared to the first learning stage–compare Fig B and Table B in S1 File.

The demonstrator was also rated as being liked more during high as compared to low affect sharing (2-way suggestion x order ANOVA: main effect of suggestion: $F(1, 37) = 38.71$, $p < .001$, $\eta_p^2 = 0.51$).

Spearman correlations showed that self-related and other-related unpleasantness ratings were correlated in the high ($r_s(34) = .51$; $p = .002$) but not in the low affect sharing condition ($r_s(34) = -.17$; $p = .310$), and that this difference between correlation coefficients was significant ($\Delta r_s = 0.680$, $z = 3.20$, $p = 0.001$). The increase in unpleasantness ratings in the high compared to the low affect sharing condition was also correlated between self and other ($r_s(34) = .34$; $p = .044$).

In line with these results, a larger number of participants reported vicarious somatosensory sensations, such as tingling, in their own body while watching the demonstrator receiving shocks in the high affect sharing condition (N = 28, for 6 of them painful, for 22 of them localized in the right arm) as compared to the low affect sharing condition (N = 3, for none of them painful, for one of them located in the right arm, McNemar test $p < .001$).

Ratings regarding declarative memory of the CS-US contingency demonstrated that participants did remember which colored square (CS) predicted a shock to the demonstrator (US), and this knowledge was not affected by the suggestions (see Fig 4B; 3-way order x suggestion x CS ANOVA: significant effect for CS, $F(1, 37) = 62.49$, $p < .001$, $\eta_p^2 = 0.63$, no effect of suggestion, $F(1, 37) = 1.21$, $p = .279$, $\eta_p^2 = 0.03$, or CS x suggestion interaction, $F(1, 37) = 0.65$, $p = .427$, $\eta_p^2 = 0.02$). There was only a significant suggestion-by-order interaction, $F(1,37) = 8.50$, $p = .006$, $\eta_p^2 = 0.19$, indicating that participants remembered fewer shocks to the demonstrator in the second compared to the first learning stage–compare Fig B and Table B in S1 File.

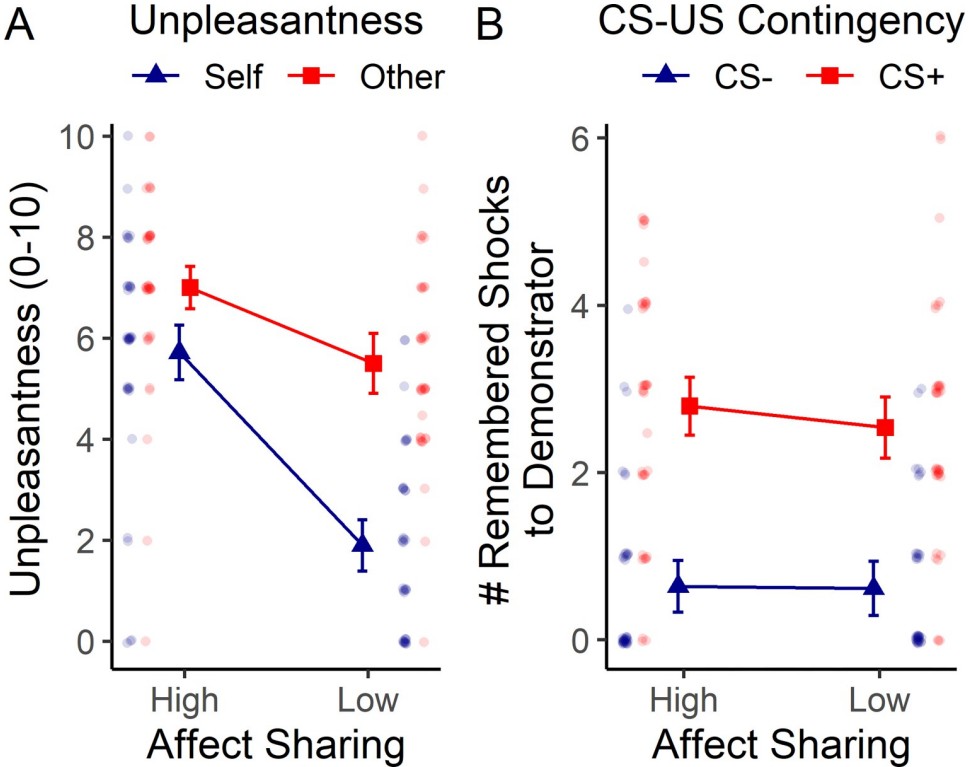

**Fig 4. Results of the post-hypnosis interview showing the effects of hypnotic suggestions for high / low affect sharing on ratings of affect sharing, emotion recognition and declarative memory of the contingency between CS (colored square) and US (shock / no shock to demonstrator).** a) Participants rated how unpleasant the shocks were for the demonstrator in the video (emotion recognition–blue / dark lines) and how unpleasant it was for themselves to watch the shocks being delivered to the demonstrator (affect sharing–red / bright lines). b) Participants indicated how many shocks the demonstrator had received following the CS+ and CS-, respectively. Error bars reflect 95% confidence intervals corrected for within-subject designs (see Methods). CS–conditioned stimulus; US–unconditioned stimulus. Results of individual participants are shown laterally as semi-transparent dots.

By calculating the absolute difference between actual and remembered number of shocks delivered to the demonstrator, we confirmed that accuracy of memory was not significantly higher for the high compared to the low affect sharing condition (see Table B and Fig A in S1 File).

## Skin conductance response

A 3-way (order x suggestion x CS) ANOVA including SCR to the CS (i.e., the colored square) as dependent variable revealed no significant effects, apart from a suggestion x order interaction, $F(1,37) = 13.25$, $p = .001$, $\eta_p^2 = 0.26$ demonstrating lower SCR in the second compared to the first learning stage–see Fig 3A and Table C in S1 File. The results indicate that no conditioned fear response to the CS was apparent yet during the learning stage (see Fig 2A).

Analysis of SCR induced by the US (i.e., the facial expression and bodily response of the demonstrator) by way of a 3-way order x suggestion x US (present / absent) ANOVA obtained significant results on all effects, including suggestion, $F(1, 37) = 10.72$, $p = .002$, $\eta_p^2 = 0.23$, US, $F(1, 37) = 65.33$, $p < .001$, $\eta_p^2 = 0.64$, and the suggestion x US interaction, $F(1, 37) = 12.40$, $p = .001$, $\eta_p^2 = 0.25$). This means that participants showed an unconditioned response (UR) to seeing the demonstrator in pain versus no pain, which was larger under high as compared to low affect sharing mainly because of an increased response to the US presence, see Fig 2B.

All these effects were stronger in the group receiving the high affect sharing suggestion in round 1 (suggestion x order: $F(1, 37) = 19.94$, $p < .001$, $\eta_p^2 = 0.35$; US x order: $F(1, 37) = 4.36$, $p = .044$, $\eta_p^2 = 0.11$; suggestion x US x order: $F(1, 37) = 9.81$, $p = .003$, $\eta_p^2 = 0.21$), and this group also had overall higher SCR than the other group (order: $F(1, 37) = 5.04$, $p = .031$, $\eta_p^2 = 0.12$), as shown in Fig 3B and in Table C in S1 File. This shows that SCR were generally smaller and less influenced by suggestion, US and the suggestion-by-US interaction in round 2 compared to round 1. A direct comparison between the SCR results obtained in round 2 versus round 1 can be found in Table F in S1 File.

## Eye gaze

Average fixation times at the cue (colored square) and face area of interest (AOI) during CS and US presentation, respectively, are displayed in Fig 5. During CS presentation, participants spent more time looking at the demonstrator's face and, conversely, less time looking at the cue during high as compared to low affect sharing (demonstrated by 3-way order x suggestion x CS ANOVAs calculated separately for each AOI [cue / face]: significant suggestion effect for face, $F(1, 34) = 11.76$, $p = .002$; $\eta_p^2 = 0.26$, and cue, $F(1, 34) = 9.35$, $p = .004$, $\eta_p^2 = 0.22$). Independently of this effect of suggestion, participants also displayed longer fixation times to the face during CS+ as compared to CS- presentation, as shown in Fig 5A (3-way ANOVAs calculated separately for each AOI showing a significant effect of CS for face, $F(1, 34) = 11.21$, $p = .002$, $\eta_p^2 = 0.25$).

Analysis of eye gaze data during the US presentation obtained very similar results: Participants looked more at the face and less at the cue during high as compared to low affect sharing–see Fig 5B (3-way order x suggestion x US ANOVAs calculated separately for each AOI: significant suggestion effect for face, $F(1, 34) = 15.75$, $p < .001$, $\eta_p^2 = 0.32$, and cue, $F(1, 34) = 14.62$, $p = .001$, $\eta_p^2 = 0.30$). Similarly and independently, participants looked more at the face and less at the cue during US as compared to US absence presentations (3-way ANOVAs calculated separately for each AOI showed a significant effect of US for face, $F(1, 34) = 18.03$, $p < .001$, $\eta_p^2 = 0.35$, and cue, $F(1, 34) = 8.78$, $p = .006$, $\eta_p^2 = 0.21$). For the complete eye tracking results see Table E in S1 File.

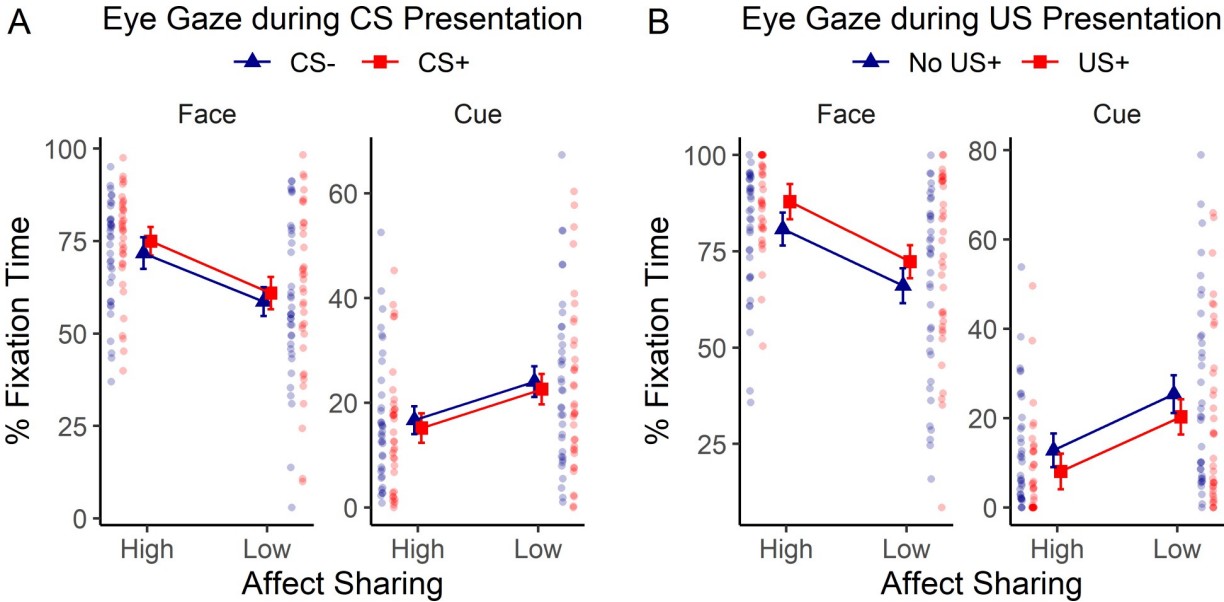

**Fig 5. Eye tracking results of the learning stage.** Average fixation times at the cue (CS) and face area of interest, expressed as % of total fixation time, recorded during the presentation of (a) the CS and (b) the US are shown separately for the high / low affect sharing condition. Error bars reflect 95% confidence intervals corrected for within-subject designs. Results of individual participants are shown laterally as semi-transparent dots.

## Tonic skin conductance level (SCL)

Changes in tonic skin conductance level (SCL) across the two learning and test stages are displayed in Fig 2D. A 3-way (stage [learning vs. test stage] x suggestion x order) ANOVA showed a significant stage x suggestion interaction ($F(1, 37) = 6.29$, $p = .017$, $\eta_p^2 = 0.15$) in the absence of any main effects (stage: $F(1, 37) = 0.15$, $p = .699$, $\eta_p^2 < 0.01$; suggestion: $F(1, 37) = 0.71$, $p = .405$, $\eta_p^2 = 0.02$), indicating that SCL tended to decrease going from learning to test stage under high affect sharing, while it showed the opposite trend under low affect sharing.

Differences between order groups: SCL in general as well as SCL in the test stage compared to the learning stage, was lower in round 2 compared to round 1 of the experiment, as indicated by a suggestion x order interaction ($F(1,37) = 32.48$, $p < .001$, $\eta_p^2 = 0.47$) and a stage x suggestion x order interaction ($F(1,37) = 22.79$, $p < .001$, $\eta_p^2 = 0.38$)–see Fig 3D.

Post-hoc 2-way (suggestion x order) ANOVAs calculated separately for each stage revealed a non-significant trend towards higher SCL under high versus low affect sharing in the learning stage (main effect of suggestion: $F(1, 37) = 3.25$, $p = .080$, $\eta_p^2 = 0.08$), which was not evident any more in the test stage ($F(1, 37) = 0.24$, $p = .631$, $\eta_p^2 = 0.01$). Post-hoc 2-way (stage x order) ANOVAs calculated separately for each suggestion revealed a non-significant trend of SCL to decrease going from learning to test stage in the high affect sharing condition (main effect of stage: $F(1, 37) = 2.87$, $p = .098$, $\eta_p^2 = 0.07$), while no such trend was evident in the low affect sharing condition ($F(1, 37) = 2.28$, $p = .139$, $\eta_p^2 = 0.06$).

## Correlations between learning and test stage

Correlations across participants among the differences in SCL, SCR and eye gaze observed between the high and low affect sharing condition are reported in Table 1 and are briefly summarized here: The only learning stage measure significantly predicting the increase in CR

**Table 1. Spearman correlations among the differences in SCL, SCR and eye gaze observed between hypnotically induced high and low affect sharing in N = 36 participants (controlling for order).**

| | SCR: CR (test) |
| --- | --- |
| Tonic SCL (learning stage) | .00 |
| Tonic SCL (test stage) | .12 |
| SCR: UR (learning stage) | -.03 |
| SCR: CR (learning stage)§ | ***.45***\*\* |
| % gaze at face vs. cue# | -.08 |

*Note.* df = 34. SCL–skin conductance level; SCR–skin conductance response; UR–unconditioned response; CR–conditioned response.

§) For the correlational analysis, only the last 3 trials of each condition are included in the calculation of the CR in the learning stage, as learning was more likely to have already occurred at this point.

#) Gaze times were averaged across all events (i.e., including CS+, CS-, US, US absence events) to represent overall gaze preference.

\*\*)p < .01.

under high versus low affect sharing in the test stage, i.e. our main effect of interest, was the increase in CR in the learning stage–see Table 1. Comparisons among the different correlations confirmed that the suggestion-induced change in CR in the test stage was significantly more correlated to the change in CR in the learning stage than to the change in tonic SCL ($\Delta r_s$ = -.45, $z$ = -2.01, $p$ = .045).), UR ($\Delta r_s$ = -.48, $z$ = -2.10, $p$ = .036) or eye gaze preference for the demonstrator's face versus the cue averaged over all CS and US events ($\Delta r_s$ = -.52, $z$ = -2.40, $p$ = .016) in the learning stage. See Tables H-J in S1 File for the complete correlation results for SCL, SCR and eye gaze.

## Discussion

Our aim was to determine whether affect sharing, an essential component of empathy, plays a role in enhancing vicarious fear learning in humans. To this end, high and low affect sharing were experimentally induced, by means of hypnotic suggestions. Using a vicarious Pavlovian fear conditioning paradigm, we assessed the effects on fear learning based on skin conductance response (SCR). Our results can be summarized in terms of three key findings.

First, SCR results of the test stage showed that participants expressed a stronger conditioned fear response to the colored square (CS) in the absence of the demonstrator when learning had occurred under high as compared to low affect sharing. This finding supports the main hypothesis motivating this research, which is that affect sharing plays a role in enhancing vicarious fear learning.

Second, results of the learning stage demonstrated that the hypnotic suggestions were effective in manipulating affect sharing. Indeed, while watching the videos in the learning stage, participants showed higher self-reported unpleasantness–a commonly used index of the affect sharing aspect of empathy, see [7, 27]–as well as a higher incidence of vicarious painful / tactile sensations in their own arm in response to seeing the demonstrator in pain (i.e., the US) during high versus low affect sharing. Skin conductance responses indicated a stronger UR during high versus low affect sharing, which was related mainly to an increased SCR amplitude to the US (i.e., the painful facial expression). There was also a non-significant trend towards higher tonic skin conductance level (SCL) throughout the learning stage during high versus low affect sharing, likely reflecting increased sympathetic arousal. This trend was not evident any more in the test stage, supporting the conclusion that the cancellation of the suggestion at the end of

the learning stage was successful. The results of the learning stage are in line with work showing that empathy for pain induced by visual scenes is associated with increased activity in brain areas involved in affective as well as somatosensory pain processing, accompanied by negative affect and sometimes even vicarious pain sensations [28, 29]. Results of the post-hypnosis interview support the validity of our experimental approach in manipulating predominantly affect sharing rather than more cognitive aspects of empathy such as mentalizing. Taken together, the findings for the learning stage document that our experimental manipulation of affect sharing was successful and had the expected effects—i.e., that it systematically manipulated the extent of affect sharing during the vicarious learning stage of the experiment.

Our third main finding was that participants spent more time watching the demonstrator's face (the US), and less the cue/colored square (the CS) area, during high as compared to low affect sharing, indicating that affect sharing is associated with increased attention toward faces and facial expressions. This finding is in line with clinical eye tracking studies, in which reduced attention toward faces has been linked to deficits in trait empathy [30–33], but also with the proposition that humans actively shift attention toward or away from affective cues in order to up-/downregulare state empathy [17, 34]. Indeed, empathy is not an automatic reflex. Instead, humans use emotion regulation strategies to approach or avoid empathic affect sharing depending on their motives in the situational context [17]. The relationship between affect sharing and attention towards emotional cues may, thus, be bi-directional. To disentangle effects of visual attention and affect sharing, future studies could employ visual bottom-up cues to draw attention towards the CS versus the US in the vicarious fear conditioning paradigm.

Overall, our findings are in line with previous studies in humans showing that vicarious fear learning could be predicted from activity in empathy-related brain regions while witnessing other people in pain [35], from trait empathy [8, 9] and from synchronicity in skin conductance activity between demonstrator and observer (which in turn has been linked to empathy; [10]), and was stronger when participants' believed the demonstrator to actually feel pain [9]. These correlational findings are consistent with the view that effects of empathy on vicarious fear learning might be graded rather than following an all-or-none principle.

The results of the correlational analyses indicate that the only significant predictor of the increase in vicarious fear learning under high versus low affect sharing observed in the test stage–i.e. our central effect of interest–was the increase in the conditioned response under high versus low affect sharing in the learning stage. Interestingly, this learning effect observed in the test stage was not significantly predicted by either the increased gaze preference towards the demonstrator's face or the increased UR to the demonstrator's pain in the learning stage during high versus low affect sharing. One possible explanation for this lack of association is that a too strong attentional focus on the US could also be detrimental to vicarious fear learning, possibly by distracting from the CS. Indeed, in a recent eye tracking study, attention toward the CS rather than the demonstrator's face was associated with stronger vicarious fear learning [8].

Contrary to our expectation, increased vicarious fear learning occurred under high versus low affect sharing in the absence of any change in declarative memory of the cue-shock (CS-US) contingency. Based on these results one might hypothesize that affect sharing can foster fear learning not (only) by strengthening associative learning of the CS-US contingency, but by increasing the threat value of the US, and maybe even more importantly, by increasing the reward value of the US absence (safety learning; [36]. On the other hand, our measure of declarative contingency memory may not have been sensitive enough to detect changes, and future studies should use more comparable measures able to separately assess US valence and CS-US association.

Because our design did not include a neutral condition without affect sharing manipulation, we cannot tell exactly to what degree the observed effects were driven by increased learning under high affect sharing versus impaired learning under low affect sharing. Olsson et al. [9] found mentalizing-based manipulations of empathy to be more effective in increasing rather than reducing vicarious fear learning compared to a control group. In contrast, our low affect sharing condition effectively abolished vicarious fear learning, and the effect size of the hypnotic manipulations was overall larger than in [9], suggesting that vicarious fear learning was probably increased as well as reduced, respectively.

Several limitations of our study are worth discussing. We chose a repeated-measures design to obtain higher statistical power with a smaller sample. Although effects of habituation / learning affecting mostly SCR / SCL data were both compensated through counterbalancing and controlled for in the statistical analysis, this reduces the comparability of our results with those of other studies using between-subjects designs, e.g. [9]. Note, however, that a roughly similar pattern of results emerged in both rounds of the experiment, as shown in Table F in S1 File.

Our correlational results have to be interpreted with caution as they are based on a small sample.

Our flat SCR electrodes are used without gel as per manufacturer instructions. Although the electrodes were attached to the skin nearly 1h before recording of the actual data occurred (see Fig 1), providing ample time for the skin under the electrode to reach a sufficient level of hydration, this might still have caused a slowly drifting signal [22]. Importantly, such slow drifts were corrected for in our analysis (see previous point) and did, thus, not impact our conclusions.

As our experimental approach manipulated predominantly but not exclusively affect sharing, leading to changes also in visual attention and more cognitive aspects of empathy such as mentalizing, further research is needed to disentangle the possible roles of these often co-occurring but distinct cognitive phenomena [7, 11, 37]. Although the validity of our manipulations was supported by the results of the post-hypnotic interviews, these qualitative data have to be interpreted with caution as results were categorized post-hoc and by raters not blind to the research questions. More generally, all self-report data were collected at the end of the experiment. While the rationale for this was to avoid influencing the fear acquisition processes during the experiment (compare [2]), it may have reduced the reliability of the self-report data.

Related to the above point, hypnotic suggestions may have affected vicarious fear learning by inducing more global changes in affective responding not relying on social processes–such as emotional numbing [15]. This interpretation is discouraged by the results of the post-hypnotic interview, in which participants frequently mentioned the demonstrator while describing the changes induced in them by the suggestions. However, the distinction between self-directed versus other-directed emotional numbing would be an interesting topic for future research on vicarious fear learning.

Our participants were informed that they could not get any shocks during the learning stage, enabling them to focus on the hypnotic suggestions without getting distracted by fear of shocks. Given that associative learning is affected by fear / stress [38, 39], our results are, thus, less comparable to similar studies, which did not give participants any information about the timing of the shocks [2]. This choice of instruction may also explain why, in contrast to other studies employing the vicarious fear conditioning paradigm [8, 40–42], our SCR data showed a significant conditioned fear response only during test, not in the learning stage, although responses were correlated between learning and test.

In addition, we pre-screened participants for hypnotic suggestibility. While this decision intended to increase sensitivity of our novel approach, the current findings may not generalize

to the general population. Given that suggestions can be effective also without the use of hypnosis, though usually to a lesser degree [14, 43, 44], future studies in the general population without hypnosis could test the generalizability of our findings.

Although we use terminology in the tradition of associative learning models throughout this text, it is important to emphasize that our results are also compatible with propositional models of observational evaluative conditioning [45]. While it is often assumed that moderators of vicarious fear learning affect the rate at which the CS-US association is learned [36], future research should also explore how and to what degree the valence of the observational US may be manipulated. Assuming that our hypnotic suggestions led to the formation of propositions about the relation between demonstrator and observer ("I do / not feel what he feels"), it is interesting to consider that these propositions had an impact although participants knew them to be the result of hypnotic imagery, and in this sense untrue.

In conclusion, by employing a novel and efficient hypnotic manipulation of affect sharing, we have demonstrated that affect sharing plays a role in enhancing vicarious fear learning as expressed through skin conductance response. Our results, thus, highlight the importance of empathy in social observational learning. Our study also points to the potential usefulness of hypnotic suggestion as an experimental tool in social neuro- and psychological science.

## Supporting information

**S1 File. Supplementary methods and results.**
(DOCX)

**S2 File. Raw data and code.**
(ZIP)

## Acknowledgments

We are grateful to Tobias Schöberl for his help with setting up the eye tracker.

## Author Contributions

**Conceptualization:** Alexa Müllner-Huber, Andreas Olsson, Claus Lamm.

**Data curation:** Alexa Müllner-Huber.

**Formal analysis:** Alexa Müllner-Huber, Ekaterina Pronizius, Lukas Lengersdorff.

**Funding acquisition:** Claus Lamm.

**Investigation:** Alexa Müllner-Huber, Lisa Anton-Boicuk.

**Methodology:** Alexa Müllner-Huber, Lisa Anton-Boicuk, Andreas Olsson, Claus Lamm.

**Project administration:** Alexa Müllner-Huber, Claus Lamm.

**Resources:** Claus Lamm.

**Software:** Alexa Müllner-Huber.

**Supervision:** Claus Lamm.

**Validation:** Claus Lamm.

**Visualization:** Alexa Müllner-Huber, Lisa Anton-Boicuk, Ekaterina Pronizius, Lukas Lengersdorff.

**Writing – original draft:** Alexa Müllner-Huber, Lisa Anton-Boicuk, Claus Lamm.

**Writing – review & editing:** Alexa Müllner-Huber, Lisa Anton-Boicuk, Ekaterina Pronizius, Lukas Lengersdorff, Andreas Olsson, Claus Lamm.

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
