## [Decision Letter · Decision Letter 0]

12 Sep 2022

PONE-D-22-08343The causal role of affect sharing in driving vicarious fear learningPLOS ONE

Dear Dr. Müllner-Huber,

Thank you for submitting your manuscript to PLOS ONE. After careful consideration, we feel that it has merit but does not fully meet PLOS ONE’s publication criteria as it currently stands. Therefore, we invite you to submit a revised version of the manuscript that addresses the points raised during the review process.

 Your manuscript has been assessed by two peer-reviewers and their reports are appended below.  The reviewers comment that  your study could be strengthened by improved reporting of the effects observed and the correlational analyses. For example, one of the reviewers suggests that the study should apply a correction for multiple testing. In addition, the reviewers comment that the discussion could be improved by including comments on the key factors driving the effects.  Could you please carefully revise the manuscript to address all comments raised? Please submit your revised manuscript by Oct 26 2022 11:59PM. If you will need more time than this to complete your revisions, please reply to this message or contact the journal office at plosone@plos.org. Please include the following items when submitting your revised manuscript:A rebuttal letter that responds to each point raised by the academic editor and reviewer(s). You should upload this letter as a separate file labeled 'Response to Reviewers'.A marked-up copy of your manuscript that highlights changes made to the original version. You should upload this as a separate file labeled 'Revised Manuscript with Track Changes'.An unmarked version of your revised paper without tracked changes. You should upload this as a separate file labeled 'Manuscript'.

We look forward to receiving your revised manuscript.

Kind regards,

Maria Elisabeth Johanna Zalm, Ph.D

Editorial Office

PLOS ONE

Journal Requirements:

2. We note that Figure 1 includes an image of a participant in the study. 

Reviewers' comments:

Reviewer's Responses to Questions

**Comments to the Author**

1. Is the manuscript technically sound, and do the data support the conclusions?

Reviewer #1: Yes

Reviewer #2: Yes

2. Has the statistical analysis been performed appropriately and rigorously? 

Reviewer #1: Yes

Reviewer #2: Yes

3. Have the authors made all data underlying the findings in their manuscript fully available?

Reviewer #1: Yes

Reviewer #2: Yes

4. Is the manuscript presented in an intelligible fashion and written in standard English?

Reviewer #1: Yes

Reviewer #2: Yes

5. Review Comments to the Author

Reviewer #1: The manuscript reports an experiment testing whether affect sharing modulates vicarious threat learning in humans. Participants undertook a vicarious Pavlovian threat conditioning procedure including a learning and a test phase under two different hypnotic suggestion conditions inducing either high or low affect sharing. During the learning phase, a higher unconditioned response (measured with skin conductance response), increased dwell time on the demonstrator’s face, and more intense self-reported unpleasantness were found when watching the demonstrator receiving an electric stimulation in the high compared to the low affect sharing condition. During the test phase, participants exhibited a greater conditioned response (operationalised as the difference in skin conductance response to the CS+ minus to the CS-) in the high relative to the low affective sharing condition, whereas no effect of the affective sharing manipulation was observed at the level of the explicit reports of the CS-US contingencies.

Overall, the study is interesting, sound, and carefully designed and conducted. I found the manuscript to be clear, the hypotheses well motivated, and the conclusions balanced, with the limitations of the study being clearly and transparently acknowledged and discussed. I would also like to commend the authors for openly sharing their data and analysis scripts in a well-documented manner. I have a few comments that could be worth considering in a revision, notably with respect to the specificity of the effects observed and the correlational analyses. I describe these comments in detail below, along with some other minor points and a suggestion.

Primary points

1) It is somewhat unclear whether the effects observed were mainly driven by (a) increased vicarious threat conditioning in the high affect sharing condition, (b) impaired vicarious threat conditioning in the low affect sharing condition, or (c) a combination of both. Even though I totally understand it is difficult to draw strong conclusions on this point without the inclusion of a control condition (e.g., hypnotic suggestion without a manipulation of affect sharing) and that adding such a condition in the current design would likely not be feasible (or advisable), I think that a discussion of the key factors driving the effects could be very beneficial to the manuscript. This could provide some deeper insights into the specificity and characteristics of the role of affect sharing in vicarious threat learning (e.g., is the influence of affect sharing an “all or none”, dichotomous effect or is it more granular, with affect sharing acting as a parametric or continuous modulator of vicarious threat learning?)

2) The fact that the hypnotic suggestions may have manipulated other cognitive factors that affect sharing is well acknowledged and discussed in the manuscript. However, I was wondering whether this manipulation could have also had an impact on other affective phenomena that do not necessarily rely on a social component. For instance, the hypnotic suggestions might have affected the relevance of emotional information in general, without being specific to its social nature. This could have in turn impacted vicarious threat learning. Although I agree this alternative explanation is rather speculative and doesn’t question the current findings, a discussion thereof could be of interest and possibly contribute to strengthening the manuscript even further.

3) It was unclear to me whether the correlational analyses performed were controlled for multiple testing. Given the number of correlations run, I would strongly recommend applying a correction for multiple testing, for instance by using false discovery rate (FDR; Benjamini & Hochberg, 1995). If the authors prefer not to apply such a correction, this should be clearly motivated and mentioned, and a clear notice warranting extra caution in the interpretation of the results from the correlation analyses should be made explicit. In any case, more information on how risks of false positives were mitigated or considered in those analyses would be good to report.

Minor points

4) On line 94: I wonder if more information could be provided on the meaning of a hypnotic suggestibility score of 7. This would help the readers who are not expert in hypnosis (like me) have a bit more context.

5) In the section “Sample and screening”, some demographic information about the characteristics of the sample (e.g., age, gender, sex, etc.) tested in the experimental study should be provided.

6) Section “Sample and screening”, ll. 105-106: I believe it would be helpful to also report the rule used for terminating data collection here (see Simmons et al., 2011).

7) Section “Test stage”, ll. 171-177: I think it would be beneficial for clarity purposes to already mention here that the test stage is performed without any electric stimulation being delivered.

8) Section “Physiological measures”, ll. 219-231: For transparency purposes, I think it would be good practice to mention in the main text (in addition to the supplementary materials) that electrocardiography was measured during the experiment.

9) Section “Eye tracking”, l. 304: I would suggest that the default settings to extract fixations be briefly described or reported.

10) Concerning Figure 2, I was not sure why the results were displayed as a function of blocks as this factor was not included in the statistical analyses. My suggestion would be that Figure 2 should specifically illustrate the results associated with the statistical analyses reported in the main text, with the current Figure 2 being moved to the supplementary materials.

11) In the legend figures (see l. 371, 398, 434, 511): The plots are described as raincloud plots. However, raincloud plots typically also include a visualisation of the distribution (i.e., “the cloud”) of the individual datapoints (and not only the individual datapoints, “the rain”; see Allen et al., 2021). Therefore, I would generally recommend avoiding referring to these plots as raincloud plots. That said, the plotting of the individual datapoints is extremely valuable and the figures are very nice.

12) In the results section, I would suggest that the specific panel of the figure in question be referred to (e.g., “as shown in Fig 4A” instead of “as shown in Fig 4”, see l. 418; see also l. 495, 516) when describing the associated results as it facilitates the identification of the relevant information in the figure.

13) In the results section, please report the degrees of freedom associated with the correlation analyses.

14) Discussion, l. 626, 632, 641: I would strongly recommend specifying “vicarious (or social) fear learning” instead of using “fear learning” only when describing the findings.

Suggestion

15) For the sake of completeness in results reporting, I would suggest that a confidence interval (e.g., 90% for tests relying on a one-sided distribution, 95% for tests relying on a two-sided distribution) around the effect size estimates be calculated and reported (see http://daniellakens.blogspot.com/2014/06/calculating-confidence-intervals-for.html). A useful R package in that regard is effectsize (https://cran.r-project.org/web/packages/effectsize/index.html;
https://easystats.github.io/effectsize/).

Signed,

Yoann Stussi

REFERENCES

Allen, M., Poggiali, D., Whitaker, K., Marshall, T. R., van Langen, J., & Kievit, R. A. (2021). Raincloud plots: A multi-platform tool for robust data visualization. Wellcome Open Research, 4, Article 63. https://doi.org/10.12688/wellcomeopenres.15191.2

Benjamini, Y., & Hochberg, Y. (1995). Controlling the false discovery rate: A practical and powerful approach to multiple testing. Journal of the Royal Statistical Society: Series B (Methodological), 57(1), 289-300. https://doi.org/10.1111/j.2517-6161.1995.tb02031.x

Simmons, J. P., Nelson, L. D., & Simonsohn, U. (2011). False-positive psychology: Undisclosed flexibility in data collection and analysis allows presenting anything as significant. Psychological Science, 22, 1359-1366. https://doi.org/10.1177/0956797611417632

Reviewer #2: This is an excellent paper that I am happy to recommend for publication pending minor revisions. I only have a few minor comments that the authors might want to take into account if asked to revise the paper.

1. Line 84: It might be good to make explicit why you expected poorer performance under low affect sharing.

2. Line 188: The fact that the results were clearly different in Round 2 than in Round 1 suggests that the attempts to avoid carry-over effects were unsuccessful. Because of these carry-over effects, would it not make sense to analyze and report only Round 1 data?

3. Line 633: It was not entirely clear to me what you mean with “was correlated between learning and test stage”. Please clarify.

4. Line 642: It might be good to clarify whether the lack of interaction was due to ceiling effects. The figure suggests that it was not, but still. Also, higher estimates do not necessarily mean better contingency awareness. There were four CS-US pairings so if someone gives an estimate of 6, he or she has poor memory for the pairings. For this reason, would it not be better to calculate the absolute differences between the number of actual CS-US pairings (i.e., 4 for the CS+) and the estimated number of CS-US pairings? Finally, what if you test the impact of suggestion on CS+ estimates only? Is this effect significant?

5. Discussion: Nothing is said about the theoretical implications of the results. In a way, I am happy about this because these data (like most other data) do not differentiate between different accounts of observational fear conditioning (e.g., association formation vs. propositional theories). The results as such are interesting because they reveal an important moderator of the effect. Nonetheless, it might be worthwhile to briefly discuss the results in relation to these models.

Signed,

Jan De Houwer

6. PLOS authors have the option to publish the peer review history of their article (what does this mean?). If published, this will include your full peer review and any attached files.

Reviewer #1: **Yes: **Yoann Stussi

Reviewer #2: **Yes: **Jan De Houwer

---

## [Author Response · Author response to Decision Letter 0]

25 Oct 2022

Dear Editor,

We have implemented the following main improvements to the manuscript to address the comments made by the Editor and the two Reviewers:

1) Methods: Additional details given as requested by the Reviewers. The faces visible in Fig. 1 have been masked to protect the individuals' identity as requested.

2) Correlational results: We have adapted our analysis approach to a more integrative approach that limits the correlational analyses to a small number of non-redundant hypothesis tests – see Table 1 and our response to comment #3 of Reviewer 1 in the Response to Reviewers. 

3) Results of an alternative index of declarative memory of the CS-US contingency – as suggested by comment #4 of Reviewer 2 – are reported in the main text and are shown in Table B and new Figure A in the Supplementary Material (S1 File), and the limits of this measure are discussed. 

4) Figure 2 showing SCR results has been adjusted to correspond to the statistical analysis (factor “block” removed), while the original Figure was moved to the Supplementary Material (now Figure D in S1 File). 

5) Discussion: We provide a more detailed discussion of possible underlying mechanisms driving the observed effects, possible limitations and implications for theoretical models of vicarious fear learning. 

We are convinced our thorough revision addressed any concerns or questions in a satisfactory way; we think the paper has greatly benefited from the peer review process, and that it will make a robust and important contribution to the literature.

Please find our detailed responses to all reviewer comments below.

Reviewer #1: ==================

Primary points:

1) It is somewhat unclear whether the effects observed were mainly driven by (a) increased vicarious threat conditioning in the high affect sharing condition, (b) impaired vicarious threat conditioning in the low affect sharing condition, or (c) a combination of both. Even though I totally understand it is difficult to draw strong conclusions on this point without the inclusion of a control condition (e.g., hypnotic suggestion without a manipulation of affect sharing) and that adding such a condition in the current design would likely not be feasible (or advisable), I think that a discussion of the key factors driving the effects could be very beneficial to the manuscript. This could provide some deeper insights into the specificity and characteristics of the role of affect sharing in vicarious threat learning (e.g., is the influence of affect sharing an “all or none”, dichotomous effect or is it more granular, with affect sharing acting as a parametric or continuous modulator of vicarious threat learning?)

We agree that these important issues should be discussed and have accordingly extended / amended the following passages in the Discussion:

ll. 657-664: “Overall, our findings are in line with previous studies in humans showing that vicarious fear learning could be predicted from activity in empathy-related brain regions while witnessing other people in pain (35), from trait empathy (8, 9) and from synchronicity in skin conductance activity between demonstrator and observer (which in turn has been linked to empathy; (10)), and was stronger when participants’ believed the demonstrator to actually feel pain (9). These correlational findings are consistent with the view that effects of empathy on vicarious fear learning might be graded rather than following an all-or-none principle.”

ll. 687-695: “Because our design did not include a neutral condition without affect sharing manipulation, we cannot tell exactly to what degree the observed effects were driven by increased learning under high affect sharing versus impaired learning under low affect sharing. Olsson et al. (2016) found mentalizing-based manipulations of empathy to be more effective in increasing rather than reducing vicarious fear learning compared to a control group. In contrast, our low affect sharing condition effectively abolished vicarious fear learning, and the effect size of the hypnotic manipulations was overall larger than in Olsson et al. (2016), suggesting that vicarious fear learning was probably increased as well as reduced, respectively.”

We have also included the following result from the post-hypnotic interview in the results section given its relevance to this point:

ll. 436-437: “In their quantitative ratings, participants described both hypnotic suggestions as equally “effective” (see Table B in S1 File).”

2) The fact that the hypnotic suggestions may have manipulated other cognitive factors than affect sharing is well acknowledged and discussed in the manuscript. However, I was wondering whether this manipulation could have also had an impact on other affective phenomena that do not necessarily rely on a social component. For instance, the hypnotic suggestions might have affected the relevance of emotional information in general, without being specific to its social nature. This could have in turn impacted vicarious threat learning. Although I agree this alternative explanation is rather speculative and doesn’t question the current findings, a discussion thereof could be of interest and possibly contribute to strengthening the manuscript even further.

We have added a discussion of this interesting point to the section on limitations: 

ll. 722-728: “Related to the above point, hypnotic suggestions may have affected vicarious fear learning by inducing more global changes in affective responding not relying on social processes – such as emotional numbing. This interpretation is discouraged by the results of the post-hypnotic interview, in which participants frequently mentioned the demonstrator while describing the changes induced in them by the suggestions. However, the distinction between self-directed versus other-directed emotional numbing would be an interesting topic for future research on vicarious fear learning.”

3) It was unclear to me whether the correlational analyses performed were controlled for multiple testing. Given the number of correlations run, I would strongly recommend applying a correction for multiple testing, for instance by using false discovery rate (FDR; Benjamini & Hochberg, 1995). If the authors prefer not to apply such a correction, this should be clearly motivated and mentioned, and a clear notice warranting extra caution in the interpretation of the results from the correlation analyses should be made explicit. In any case, more information on how risks of false positives were mitigated or considered in those analyses would be good to report.

We agree that a correction for multiple comparisons would have been in order for the large number of correlations presented in our previous Table 1, especially because it contained four measures of eye gaze highly redundant with each other. Given the mostly moderate correlations observed in previous similar studies (Olsson et al., 2016; Kleberg et al., 2015), we have adapted our analysis approach to a more integrative approach that limits the correlational analyses to a small number of non-redundant hypothesis tests, which do not require any correction for multiple comparisons because no hypothesis was tested multiple times (Rubin, 2021). To this end, Table 1 (see l. 596) has been reduced to show only correlations with our main outcome measure, i.e. the increased conditioned fear response under high versus low affect sharing in the test stage. The correlational analysis now serves to assess which of the physiological changes observed in the learning stage (SCL, SCR, eye gaze) can predict this main outcome. The four redundant eye gaze variables were replaced by a single composite eye gaze index – see details below. As these different learning stage indices have different theoretical meanings, Table 1 does not contain any multiple tests of a single hypothesis. The Methods and Results sections have been adjusted as follows: 

ll. 349-355: Methods: statistical analysis: “To avoid type I error inflation caused by multiple redundant eye gaze variables, a composite eye gaze index was calculated by averaging the % fixation time at the demonstrator’s face versus the cue across all CS and US events, separately for each suggestion condition. Only this composite index, which represents the only significant change in eye gaze induced by suggestions (see Results section and Table E in S1 File) was included in the correlation analysis, to assess whether the magnitude of changes in eye gaze correlated with SCR measures.”

ll. 575-593: Results: Correlations between learning and test stage: “Correlations across participants among the differences in SCL, SCR and eye gaze observed between the high and low affect sharing condition are reported in Table 1 and are briefly summarized here: The only learning stage measure significantly predicting the increase in CR under high versus low affect sharing in the test stage, i.e. our main effect of interest, was the increase in CR in the learning stage – see Table 1. Comparisons among the different correlations confirmed that the suggestion-induced change in CR in the test stage was significantly more correlated to the change in CR in the learning stage than to the change in tonic SCL (Δrs = -.45, z = -2.01, p = .045). ), UR (Δrs = -.48, z = -2.10, p = .036) or eye gaze preference for the demonstrator’s face versus the cue averaged over all CS and US events (Δrs = -.52, z = -2.40, p = .016) in the learning stage. See Table H-J in S1 File for the complete correlation results for SCL, SCR and eye gaze.” 

ll. 704-705: Discussion of limitations: “Our correlational results have to be interpreted with caution as they are based on a small sample.”

Minor points:

4) On line 94: I wonder if more information could be provided on the meaning of a hypnotic suggestibility score of 7. This would help the readers who are not expert in hypnosis (like me) have a bit more context.

Has been rephrased for greater clarity:

ll. 94-103: Methods: Sample and screening: “N=410 first- and second-year Bachelor Psychology students of the University of Vienna were screened for hypnotic suggestibility using the Harvard Group Scale of Hypnotic Susceptibility (HGSHS:A) (18). Participants were recruited from this pool if they had a hypnotic suggestibility score of at least 7 on a scale ranging from 0-12, in line with previous hypnosis studies, e.g. (12). This criterion was met by approximately one third of the student population. Additional exclusion criteria were a history of chronic pain, acute pain on the day of the experiment, previous experience with experimental electrical pain stimuli, a history of substance abuse or psychiatric or severe organic disease, and chronic or recent use of opiates.”

5) In the section “Sample and screening”, some demographic information about the characteristics of the sample (e.g., age, gender, sex, etc.) tested in the experimental study should be provided.

ll. 115-119: Methods: Sample and screening: additional information on gender and age provided: 

“Another three participants were excluded from the analysis of eye gaze data due to missing gaze data (N=2; see details below) or technical problems with the eye tracker (N=1), resulting in a final sample size of 39 participants (30 female, 9 male; mean age 20.0 years, range 18-24 years) for self-report and SCR, and 36 participants (28 female, 8 male) for the eye gaze results.”

6) Section “Sample and screening”, ll. 105-106: I believe it would be helpful to also report the rule used for terminating data collection here (see Simmons et al., 2011).

ll. 104-112: Sample and screening: This information has been added:

“Based on the medium effect size of Cohen’s d=0.50 observed in Olsson et al. (9), we conducted an a priori power analysis to detect a medium effect size (Cohen’s f = 0.25) with the conventional power of 80% for our central effect of interest, i.e. the CS-by-suggestion interaction observed in a 3-way mixed ANOVA including the repeated-measures factors CS and suggestion and the between-subjects factor presentation order. This yielded a minimum sample size of N=34 participants using a within-subject design. Due to delays in the technical data inspection, data collection was terminated once we had collected 36 participants with technically valid data on all measures.”

7) Section “Test stage”, ll. 171-177: I think it would be beneficial for clarity purposes to already mention here that the test stage is performed without any electric stimulation being delivered.

ll. 182-184: Methods: Test stage: This information has been added:

“In line with standard procedures (2) aimed at ensuring that only indirect, vicarious learning is measured, the participant never actually received any shocks during the entire test stage.”

8) Section “Physiological measures”, ll. 219-231: For transparency purposes, I think it would be good practice to mention in the main text (in addition to the supplementary materials) that electrocardiography was measured during the experiment.

ll. 235-237: Methods: Physiological measures: Information has been added to the main text:

“Electrocardiogram (ECG, outside scope of present paper), skin conductance and eye movements were recorded throughout the paradigm.”

9) Section “Eye tracking”, l. 304: I would suggest that the default settings to extract fixations be briefly described or reported.

ll. 315-320: Methods: eye tracking: Information has been added:

“The Eyelink 1000 Plus online parser was used with default settings to extract fixations, saccades and blinks from the eye movement data (“cognitive configuration” for Remote Mode: use gaze data to compute velocity; velocity threshold of saccade detection: 40°/sec; acceleration threshold of saccade detector: 80000°/sec/sec; minimum motion out of fixation before saccade onset allowed: 0.2°; maximum pursuit velocity accommodation by the saccade detector: 60°/sec; fixation update interval: 50 msec).”

10) Concerning Figure 2, I was not sure why the results were displayed as a function of blocks as this factor was not included in the statistical analyses. My suggestion would be that Figure 2 should specifically illustrate the results associated with the statistical analyses reported in the main text, with the current Figure 2 being moved to the supplementary materials.

We agree and have altered Figure 2 accordingly. Results displayed as a function of blocks can now be seen in Figure D in the Supplementary Material. 

11) In the legend figures (see l. 371, 398, 434, 511): The plots are described as raincloud plots. However, raincloud plots typically also include a visualisation of the distribution (i.e., “the cloud”) of the individual datapoints (and not only the individual datapoints, “the rain”; see Allen et al., 2021). Therefore, I would generally recommend avoiding referring to these plots as raincloud plots. That said, the plotting of the individual datapoints is extremely valuable and the figures are very nice.

Any reference to “raincloud plots” has been removed from the main text and from the Supplementary Material. 

12) In the results section, I would suggest that the specific panel of the figure in question be referred to (e.g., “as shown in Fig 4A” instead of “as shown in Fig 4”, see l. 418; see also l. 495, 516) when describing the associated results as it facilitates the identification of the relevant information in the figure.

Has been adjusted throughout the results section.

13) In the results section, please report the degrees of freedom associated with the correlation analyses.

Degrees of freedom have been added to the results section of the main text:

ll. 470-475: Results: “Spearman correlations showed that self-related and other-related unpleasantness ratings were correlated in the high (rs(34) = .51; p = .002) but not in the low affect sharing condition (rs(34) = -.17; p = .310), and that this difference between correlation coefficients was significant (Δrs = 0.680, z = 3.20, p = 0.001). The increase in unpleasantness ratings in the high compared to the low affect sharing condition was also correlated between self and other (rs(34) = .34; p = .044).”

l. 599: df have been added in the notes under Table 1.

Df have also been added the Supplementary Material in the notes under Tables H-J. 

14) Discussion, l. 626, 632, 641: I would strongly recommend specifying “vicarious (or social) fear learning” instead of using “fear learning” only when describing the findings.

Has been added:

ll. 657-662: “Overall, our findings are in line with previous studies in humans showing that vicarious fear learning could be predicted from activity in empathy-related brain regions while witnessing other people in pain (35), from trait empathy (8, 9) and from synchronicity in skin conductance activity between demonstrator and observer (which in turn has been linked to empathy; (10)), and was stronger when participants’ believed the demonstrator to actually feel pain (9).”

ll. 665-669: “The results of the correlational analyses indicate that the only significant predictor of the increase in vicarious fear learning under high versus low affect sharing observed in the test stage – i.e. our central effect of interest – was the increase in the conditioned response under high versus low affect sharing in the learning stage.”

ll. 677-679: “Contrary to our expectation, increased vicarious fear learning occurred under high versus low affect sharing in the absence of any change in declarative memory of the cue-shock (CS-US) contingency.”

15) For the sake of completeness in results reporting, I would suggest that a confidence interval (e.g., 90% for tests relying on a one-sided distribution, 95% for tests relying on a two-sided distribution) around the effect size estimates be calculated and reported (see http://daniellakens.blogspot.com/2014/06/calculating-confidence-intervals-for.html). A useful R package in that regard is effectsize (https://cran.r-project.org/web/packages/effectsize/index.html;
https://easystats.github.io/effectsize/).

Confidence intervals around the effect size estimates have been added to all ANOVA results reported in the Supplementary Material. 

Reviewer #2: ======================

1. Line 84: It might be good to make explicit why you expected poorer performance under low affect sharing.

ll. 82-87: Introduction: rephrased for greater clarity: “Finally, as participants may learn the cue-shock contingency even if they are not afraid of the shocks, declarative memory of the cue-shock contingency was tested at the end of the experiment as an alternative learning measure less reliant on fear. We expected better performance under high compared to low affect sharing, based on the assumption that high affect sharing should foster associative learning of the cue-shock contingency.”

2. Line 188: The fact that the results were clearly different in Round 2 than in Round 1 suggests that the attempts to avoid carry-over effects were unsuccessful. Because of these carry-over effects, would it not make sense to analyze and report only Round 1 data?

To reduce learning effects across rounds, we used a different CS colour pair and a different demonstrator in round 2 compared to round 1 (counterbalanced across participants). Nevertheless, and in line with known habituation effects of skin conductance, SCR and SCL tended to decrease from round 1 to round 2, as rightly pointed out by the Reviewer. In consequence, some (but not all) experimental effects were weaker in round 2 compared to round 1. This is summarized in Table F in the Supplementary Material, which shows SCR / SCL results calculated separately for each round, and also a comparison between both. Note how the learned fear response in the test stage is significant in both rounds, with no significant difference between rounds (as indicated in Table F by a significant main effect of CS in both test stages and a non-significant comparison between both). In addition, our main effect of interest – higher fear learning in the test stage under high versus low affect sharing (indicated by the CS x order interaction in Table F because these are between-subjects instead of within-subjects comparisons) – is not less significant in round 2 compared to round 1, indicating that our learning paradigm and our experimental manipulations were still effective in round 2 in spite of habituation effects. 

In conclusion, although some effects (notably the unconditioned response – compare Table F) may have been rendered slightly less significant by habituation effects, we feel it is justified to analyze results across both rounds because the pattern of our effects of interest is consistent across rounds, and because using the data from round 1 only (and, thus, conducting between-groups instead of repeated-measures analyses) would mean using a sample underpowered to answer our research questions.

We have adjusted the Discussion section on limitations to draw more attention to the Supplementary Results provided on this topic: 

ll. 696-703: “We chose a repeated-measures design to obtain higher statistical power with a smaller sample. Although effects of habituation / learning affecting mostly SCR / SCL data were both compensated through counterbalancing and controlled for in the statistical analysis, this reduces the comparability of our results with those of other studies using between-subjects designs, e.g. (9). Note, however, that a roughly similar pattern of results emerged in both rounds of the experiment, as shown in Table F in S1 File.” 

3. Line 633: It was not entirely clear to me what you mean with “was correlated between learning and test stage”. Please clarify.

ll. 665-672: Discussion: Text has been rephrased for greater clarity:

“The results of the correlational analyses indicate that the only significant predictor of the increase in vicarious fear learning under high versus low affect sharing in the test stage – i.e. our central effect of interest – was the increase in vicarious fear learning under high versus low affect sharing in the learning stage. Interestingly, this learning effect observed in the test stage was not significantly predicted by either the increased gaze preference towards the demonstrator’s face or the increased UR to the demonstrator’s pain in the learning stage during high versus low affect sharing.”

4. Line 642: It might be good to clarify whether the lack of interaction was due to ceiling effects. The figure suggests that it was not, but still. Also, higher estimates do not necessarily mean better contingency awareness. There were four CS-US pairings so if someone gives an estimate of 6, he or she has poor memory for the pairings. For this reason, would it not be better to calculate the absolute differences between the number of actual CS-US pairings (i.e., 4 for the CS+) and the estimated number of CS-US pairings? Finally, what if you test the impact of suggestion on CS+ estimates only? Is this effect significant?

We thank the Reviewer for these excellent suggestions. We have calculated the absolute difference between actual and estimated number of CS-US pairings as an additional measure of contingency memory as suggested by the Reviewer. To avoid a too lengthy manuscript, results of this analysis are shown in Table B and in Figure A of the Supplementary Material, and the following paragraph has been added to the main text:

ll. 492-495: Results: “By calculating the absolute difference between actual and remembered number of shocks delivered to the demonstrator, we confirmed that accuracy of memory was not significantly higher for the high compared to the low affect sharing condition (see Table B and Fig A in S1 File).” 

Although we agree with the Reviewer that no apparent ceiling / floor effects are evident in Figure 4B and Figure A (S1 File), the measure may still have had insufficient sensitivity, which has been added to the Discussion:

ll. 677-686: Discussion: “Contrary to our expectation, increased vicarious fear learning occurred under high versus low affect sharing in the absence of any change in declarative memory of the cue-shock (CS-US) contingency. Based on these results one might hypothesize that affect sharing can foster fear learning not (only) by strengthening associative learning of the CS-US contingency, but by increasing the threat value of the US, and maybe even more importantly, by increasing the reward value of the US absence (safety learning; Olsson et al., 2020). On the other hand, our measure of declarative contingency memory may not have been sensitive enough to detect changes, and future studies should use more comparable measures able to separately assess US valence and CS-US association.” 

5. Discussion: Nothing is said about the theoretical implications of the results. In a way, I am happy about this because these data (like most other data) do not differentiate between different accounts of observational fear conditioning (e.g., association formation vs. propositional theories). The results as such are interesting because they reveal an important moderator of the effect. Nonetheless, it might be worthwhile to briefly discuss the results in relation to these models.

We have added a paragraph at the end of the Discussion to relate our findings to existing theoretical models and to mention important future research directions:

ll. 747-756: Discussion: “Although we use terminology in the tradition of associative learning models throughout this text, it is important to emphasize that our results are also compatible with propositional models of observational evaluative conditioning (Kasran et al., 2022). While it is often assumed that moderators of vicarious fear learning affect the rate at which the CS-US association is learned (Olsson et al., 2020), future research should also explore how and to what degree the valence of the observational US may be manipulated. Assuming that our hypnotic suggestions led to the formation of propositions about the relation between demonstrator and observer (“I do / not feel what he feels”), it is interesting to consider that these propositions had an impact although participants knew them to be the result of hypnotic imagery, and in this sense untrue.” 

References:

Rubin M. When to adjust alpha during multiple testing: a consideration of disjunction, conjunction, and individual testing. Synthese 2021;199:10969 – 11000.

---

## [Decision Letter · Decision Letter 1]

4 Nov 2022

The causal role of affect sharing in driving vicarious fear learning

PONE-D-22-08343R1

Dear Dr. Müllner-Huber,

We’re pleased to inform you that your manuscript has been judged scientifically suitable for publication and will be formally accepted for publication once it meets all outstanding technical requirements.

Kind regards,

Thiago P. Fernandes, PhD

Academic Editor

PLOS ONE

Reviewers' comments:

Reviewer's Responses to Questions

**Comments to the Author**

1. If the authors have adequately addressed your comments raised in a previous round of review and you feel that this manuscript is now acceptable for publication, you may indicate that here to bypass the “Comments to the Author” section, enter your conflict of interest statement in the “Confidential to Editor” section, and submit your "Accept" recommendation.

Reviewer #1: All comments have been addressed

Reviewer #2: All comments have been addressed

2. Is the manuscript technically sound, and do the data support the conclusions?

Reviewer #1: Yes

Reviewer #2: Yes

3. Has the statistical analysis been performed appropriately and rigorously? 

Reviewer #1: Yes

Reviewer #2: Yes

4. Have the authors made all data underlying the findings in their manuscript fully available?

Reviewer #1: Yes

Reviewer #2: Yes

5. Is the manuscript presented in an intelligible fashion and written in standard English?

Reviewer #1: Yes

Reviewer #2: Yes

6. Review Comments to the Author

Reviewer #1: The authors have provided a thorough revision of their manuscript. They have carefully and convincingly addressed all my previous comments. Overall, the revision contributed to strengthening the manuscript and I have no further comment. I believe the paper is ready for publication and will make a valuable contribution to the field of social threat learning in humans.

Signed,

Yoann Stussi

Reviewer #2: (No Response)

7. PLOS authors have the option to publish the peer review history of their article (what does this mean?). If published, this will include your full peer review and any attached files.

Reviewer #1: **Yes: **Yoann Stussi

Reviewer #2: **Yes: **Jan De Houwer

---

## [Editor Report · Acceptance letter]

9 Nov 2022

PONE-D-22-08343R1 

The causal role of affect sharing in driving vicarious fear learning 

Dear Dr. Müllner-Huber:

I'm pleased to inform you that your manuscript has been deemed suitable for publication in PLOS ONE. Congratulations! Your manuscript is now with our production department. 

Kind regards, 

on behalf of

Dr. Thiago P. Fernandes 

Academic Editor

PLOS ONE